# The Impact of the COVID-19 Pandemic on the Food Security of UK Adults Aged 20–65 Years (COVID-19 Food Security and Dietary Assessment Study)

**DOI:** 10.3390/nu14235078

**Published:** 2022-11-29

**Authors:** Michelle Thomas, Elizabeth Eveleigh, Zeynep Vural, Peter Rose, Amanda Avery, Lisa Coneyworth, Simon Welham

**Affiliations:** Nutrition and Dietetics, Division of Food Science, School of Biosciences, University of Nottingham; Sutton Bonington LE12 5RD, UK

**Keywords:** food security, micronutrients, diet, high income households, COVID-19, national lockdown, United Kingdom

## Abstract

The first UK lockdown greatly impacted the food security status of UK adults. This study set out to establish if food procurement was adapted differently for different income groups and if this impacted dietary intakes disproportionately. Adults (*n* = 515) aged 20–65 years participated in an online survey with 56 completing a 3–4 day diet diary. Food availability was a significant factor in the experience of food insecurity. Similar proportions of food secure and food insecure adapted food spend during lockdown, spending similar amounts. Food insecure (*n* = 85, 18.3%) had a 10.5% lower income and the money spent on food required a greater proportion of income. Access to food was the biggest driver of food insecurity but monetary constraint was a factor for the lowest income group. The relative risk of food insecurity increased by 0.07-fold for every 1% increase in the proportion of income spent on food above 10%. Micronutrient intakes were low compared to the reference nutrient intake (RNI) for most females, with riboflavin being 36% lower in food insecure groups (*p* = 0.03), whilst vitamin B12 was 56% lower (*p* = 0.057) and iodine 53.6% lower (*p* = 0.257) these were not significant. Coping strategies adopted by food insecure groups included altering the quantity and variety of fruit and vegetables which may have contributed to the differences in micronutrients.

## 1. Introduction

The virus “severe acute respiratory syndrome coronavirus 2” which causes the coronavirus disease (referred to in the study from here as COVID-19) led to the United Kingdom’s (UK) first ‘lockdown’ on the 23 March 2020 until 10 May 2020, when restrictions were eased [1]. In the first 3 weeks of this lockdown more than 3 million people reported that they had gone hungry [2]. Additionally, all non-essential shops, schools (to all children whose caregivers did not work in key sector organizations), the accommodation and food sector, the arts, entertainment and recreation sector were closed to halt the spread of the disease [3]. Concerns of having to remain in the home or isolate for long periods resulted in increased demand for food from supermarkets for consumption in the home. What ensued was empty supermarket shelves, with staple food items such as pasta, rice and flour being in short supply. Whilst the UK food supply chains had adequate produce, the change in consumer behavior caused shortages as retailers were unable to keep up with the increased demand. This negatively impacted food supply to the consumer via the retail sector [4].

With the spread of the disease and closure of many sectors of UK industries, the Government introduced ‘The Coronavirus Job Retention Scheme’ (1 March 2020–30 September 2021). The aims of the scheme were to reduce the burden on Social Security and enable employers to retain staff and pay up to 80% of employee’s usual monthly salary (capped at £2500 per month). For some, this resulted in a loss of income of 20% or more (depending on baseline income), for others, their employers topped up their wages to the full amount. Self-employed individuals received some support but this was only available to individuals if they had submitted a Self-Assessment tax return for the year 2018–2019 and had traded in the year 2019–2020 [5]. For others there was a complete loss of employment and this was particularly pronounced for younger people, who were less likely to be furloughed than those aged over 65 who were still in employment [6]. The consequence of loss of income and or employment was evident from the sharp increase in claims for Universal Credit (UC) during April and May 2020. The typical number of claims prior to COVID-19 was ~200,000 per month [7]. This increased to 1.2 million in April and 1.3 million in May 2020.

The COVID-19 pandemic had an impact on food and nutrition security both directly (food shortages) and indirectly (loss of income/purchasing power [3]). The baseline situation of communities, households and individuals (i.e., low income, living in deprived regions and limited capacity for working at home) was found to be a risk factor in the experience of food insecurity during COVID-19 [3]. As supermarkets struggled to keep pace with demand, individuals became more likely to over-purchase (defined as buying more than necessary to sustain routine practices within a household) food, toiletries and pharmacy products [8]. These individuals were more likely to be younger, female, have children living at home, and have a high income or conversely, suffering from a loss of income [8]. It has been suggested that loss of income was a factor in panic buying (defined as overbuying despite sufficient commodities within the supply chain) as individuals were concerned about future scarcity [8]. For many with a higher income, this afforded the opportunity to buy extra, to the detriment of lower income groups who did not have the equivalent purchasing power [9]. As a consequence of these factors, principle difficulties with the food supply were more a result of the many buying a little extra in times of uncertainty than from the few purchasing in excess [8].

The COVID-19 pandemic brought into sharp focus the concerns for population groups with increased vulnerability to the experience of food insecurity. Loss of income or employment increased demand for aid from food banks and the elevated general demand for food from the retail sector compromised the food security (having sustained physical and financial access to a safe and healthy varied diet that meets nutritional requirements) [10] of low income groups resulting in many experiencing food insecurity for the first time [9]. In the UK, these were typically young adults (18–24), households with children, minority ethnic groups, individuals with disabilities, and low income and unemployed households [9].

Food security can be compromised for a range of reasons which impact an individual’s or a household’s ability to procure food consistently to meet their dietary, nutritional and social requirements [11]. Experience of food insecurity and poor diet directly contribute to increased incidence of disease and lower life expectancy [12,13,14]. In addition, the experience of food insecurity can make it hard for adults to maintain stable employment [9]. A low income does not always equate to being food insecure and similarly, the anxiety around food availability is not the sole preserve of those on lower incomes [15]. The response to concerns of food security frequently results in similar dietary choices independent of background [16,17], with individuals and households experiencing food insecurity selecting high energy dense, nutrient poor, cheaper foods [18,19], which may be perceived as better value for money and more accessible under the circumstances faced [17]. Fresh fruit and vegetables are often sacrificed at the expense of high fat, high sugar alternatives such as crisps and biscuits [16,18]. Dairy products and protein rich foods may also be limited [20]. These dietary choices may negatively affect long-term health and well-being.

In this study, we assessed the impact of social isolation and movement restriction on food availability and food security in UK adults during the first COVID-19 lockdown period hypothesising that the nutrient profile of diets would change detrimentally during lockdown, resulting in the consumption of a more energy dense diet and that a significant number of people would self-classify as food insecure. We additionally set out to understand people’s perceptions of their food security, their food purchasing choices and if these were reflected in their actual intakes.

## 2. Materials and Methods

This paper details the findings from a cross-sectional study which took place during the first UK COVID-19 pandemic lockdown between 6 May and 10 July 2020 for adults aged between 20 and 65 years who were not in education. An online survey was designed to collect general demographic information (age, gender, ethnicity, highest level of education attained, employment status, post-code), self-reported weight and height, household characteristics, whether following government guidelines on isolating and shielding, indicators of food purchasing behaviour, food security and dietary change.

### 2.1. Participant Recruitment

Participants were recruited to the study via social media platforms (Twitter, Facebook), radio appearances, the University communication team and word of mouth. All participants were provided with information about the study and asked to give consent before completing the survey. The study was approved by the University of Nottingham’s Faculty of Medicine and Health Sciences Research Ethics Committee (Ethics Reference Number 01-0420). This research project was completed in accordance with the declaration of Helsinki and recent alterations.

### 2.2. Equivalised Income and Income Quintiles

Questions were adapted from the National Diet and Nutrition Survey (NDNS) [21] to determine the level of household income. Participants were asked to select an income bracket and the midpoint value of each income bracket was used in the calculation of equivalised household income (EHI), along with a household size score, adapted from the McClements scale where a value was assigned to each of the adults living in a household and, where applicable, to the children based on their ages [22,23,24] Household income was divided by the adapted McClements score to determine equivalised household income.

Participants were excluded from the analysis if the household size was greater than 1.5 times the inter-quartile range. As a consequence, 3 participants were excluded from the analysis. Two participants listed they had 11 children within the same age bracket and one participant was excluded as they listed, they had 11 children and 11 adults in each of the age brackets.

Income quintiles (IQ) were determined by splitting the equivalised income during lockdown into five percentiles as follows IQ1 (*n* = 98; 20.9%; <£25,700.47), IQ2 (*n* = 90; 19.1%; £25,700.47–£39,643.18), IQ3 (*n* = 99; 21.1%; £39,643.18–£53,277.84), IQ4 (*n* = 84; 17.9%; £53,277.84–£75,503.02), IQ5 (*n* = 99; 21.1%; >£75,503.02).

### 2.3. Food Security Measures

Food security was assessed with nine questions adapted from the Household Food Insecurity Access Scale (HFIAS) [25] to determine influences of monetary resources and/or food availability over the previous 4 weeks (from the date of completing the survey) on household food security. The HFIAS assesses three different but related domains of food insecurity [25]. Positive responses across the domains indicate increasing severity of food insecurity experienced. We adapted the questions to evaluate if the experience of food insecurity was because of a lack of money or lack of food.

Domain one is concerned with anxiety/worry of running out of food and asks the question (1) “Did you worry that your household would not have enough food”. Domain two includes three questions to assess if there was a reduction in the quality and variety of the food consumed. These questions asked (2) “were you or any household member not able to eat the kinds of foods you preferred because of a lack of money or lack of food available?”, (3) “Did you or any household member have to eat a limited variety of foods due to lack of money or food available?” and (4) “Did you or any household member have to eat same food that you really did not want to eat because of lack of money or lack of food available to obtain other types of food”.

The final domain asks five questions and is concerned with reduction in the quantity of food eaten and experience of hunger. The first asks (5) “Did you or any household member have to eat a smaller meal than you felt you needed because there was not enough food?”, and the second, (6) “Did you or any household member have to eat fewer meals in a day because there was not enough food?”. Additional questions ask (7) “Was there ever no food to eat of any kind in your household because of a lack of money or lack of food available to get food?”, (8) “Did you or any household member go to sleep at night hungry because there was not enough food?” and (9) “Did you or any household member go a whole day and night without eating anything because there was not enough food” [25].

Participants were initially categorised into the food security domains of food secure or mild, moderate, or severely food insecure. Two categories were then created comprising food secure and food insecure.

### 2.4. Shopping Habits and Food Spend

Participants were asked about their food shopping behaviours before and during the first UK national lockdown in reference to where food was purchased, how and how frequently (never less than once a month, 2–3 times per month, once a month, 2–4 times per week, 5–6 times per week, once a day, prefer not to say). The following question was asked with the following options for response “Which of following best describe where you purchased foods from? (Tick all that apply)”: (1) Shop at one of the UK “Big Four” supermarkets (Tesco, Sainsbury’s, Morrisons, Asda) (2) “In person”, (3) “home delivery”, (4) “Click and Collect” (5) Other supermarket (Aldi, Lidl, Iceland, Netto). (6) “Other supermarket “(Waitrose, Marks and Spencer), (7) smaller shops (e.g., Co-op, Tesco express, Sainsbury local), (8)” Corner Shops (e.g., Happy Shopper, 7-11, Spar), (9) “Markets”, (10) Local independents (e.g., butchers, bakers, green grocers). In addition, participants were asked whether they were self-isolating or shielding and their level of vulnerability. Individuals were asked about usual eating behaviours, dietary choices, perception of how food availability had changed, and how their diet had changed during the lockdown. Food spend was estimated for each household from the mid-point of the monetary bracket per week (<£46, £47–£69, £70–£90, £91–£115, £116–£138, £139–£161, >£162) selected by participants.

### 2.5. Energy and Nutrient Intakes

Participants had the option to complete a 4-day food diary using the “Libro” app associated with professional dietary analysis software (Nutritics). Those who completed 3- or 4-day food diaries were included in the analysis (*n* = 56). We present the results for the total population of females and do not exclude non-plausible reporters due to the nature of the study assessing the impact of food insecurity on the energy and nutrient composition of the diet. The macronutrient and micronutrient composition of each participant’s diet was calculated by the Nutritics software. Analysis of the micronutrient composition of the diet and food security status was completed for the total population. Females were stratified by age as per the reference nutrient intake (RNI) categories to enable analysis of iron intakes (19–49 and 50+ years).

### 2.6. Sensitivity Analysis

The plausibility of energy intake was assessed by estimating Energy Intake: Basal Metabolic Rate (EI:BMR) ratio using the Schofield equation to estimate BMR and applying the Goldberg upper and lower and cut-off points specific to physical activity level (PAL; Appendix A) [26,27].

### 2.7. Data Analysis

Descriptive, parametric, and non-parametric analyses were performed using SPSS (IBM Corp. Released 2020. IBM SPSS Statistics for Windows, Version 27.0. Armonk, NY,: IBM Corp). Normality of the data was assessed in SPSS using Shapiro–Wilks. Parametric data are presented as means and S.E.M (unless otherwise stated), non-parametric as medians with 25th and 75th percentile (median [25th–75th percentile]). Chi-square was used for categorical variables to test the impact of income quintile on food security status. Relative risk was calculated for the experience of food insecurity according to employment type, adherence to government guidelines for movement restriction, household income quintiles, and food spend as a proportion of income. Dietary data were analysed for participants who completed 3 or 4 days of a food diary. Parametric and non-parametric tests were completed in SPSS to test for difference in dietary intake between individuals considered food secure and food insecure.

## 3. Results

This study recruited 515 participants between 20 and 65 years of age (43.3 ± 0.5 years) of which the majority were female (*n* = 435; 84%). Males (*n* = 79; 15%) had a mean age of 43.5 ± 0.6 years (Table 1). One participant did not provide their sex. The majority of participants in this study stated their ethnicity as white British (*n* = 422; (81.9%)), whilst 11.3% were white Irish or white other, 2.2% Asian, 0.4% white and black African or African, 0.2% Arab, 0.8% other and 0.2% preferred not to say. Two people did not provide details of their ethnicity. During the first lockdown, the proportion of participants in employment was 73.7% (*n* = 390) of which over half were employed full time (51.6%), 15.5% (*n* = 82) part time and 6.6% were self-employed (*n* = 35). The proportion of respondents not in paid employment was 26.4% of which retirees accounted for 6.6% (*n* = 35) and furloughed workers 9.8% (*n* = 52). Fourteen participants selected more than one option for employment type.

The study cohort was disproportionately represented by those who had successfully accessed higher education. Most participants (*n* = 405; 78.6%) had completed their education to level 6 (undergraduate degree with honours or equivalent) or above, with 35.9% having an undergraduate degree, 30.3% a post graduate degree at master level or equivalent, and 12.4% a PhD or DPhil. Only 0.4% reported having no qualifications. In the UK, by contrast, between April 2020 and March 2021, approximately 20.8% of the population reported they had a degree level qualification or above [28].

The median equivalised household income for all participants with a valid household income prior to and during lockdown (*n* = 470) decreased 5.5% from £46,969.22 [£33,783.51–£68,130.11] to £44,392.06 [£28,687.70–£61,474.59] per year. Prior to the lockdown, 81.1% of households had an income above the UK median average household income for 2020 (£29,990). This reduced to 73.8% during the first lockdown. We found 5.3% (*n* = 25) of households had an income below 60% of the UK median (£13,794.00; a level used for defining relative low income) prior to lockdown, which increased to 8.1% (*n* = 38) during the lockdown.

The largest group of households were two person (*n* = 200, 39.1%; 3, 1.5% with children) followed by 4 person (*n* = 103, 20.1%; 78, 76% with children) and 3 person (*n* = 100, 19.5%; 47, 47% with children). Single person households accounted for 14.5% (*n* = 74). Households with children comprised over a third (30.1%; *n* = 155).


**Who was at risk of the experience of food insecurity?**


### 3.1. Equivalised Household Income

Four fifths of participants in this study were food secure (81.7%). Of those who experienced some form of food insecurity (18.3%), 2.9% indicated they were severely food insecure. Participants who provided details about household income before and during lockdown (*n* = 470) were split into income quintiles (IQ; Table 1). Households in IQ1 (income < £25,700.47 per year) had the lowest proportion of food secure households (73.5%) compared to 1Q2 (83.3%), IQ3 (80.8%), IQ4 (86.9%) and IQ5, (85.9%) and the highest percentage of severely food insecure (8.2%) compared to participants in IQ2 (0%), IQ3 (2%), IQ4 (0%) and IQ5 (3.0%). Two participants who identified as severely food insecure in IQ3 (*n* = 1) and IQ5 (*n* = 1) had restricted diets due to coeliac disease. The participant in IQ3 stated,

“I follow a gluten free diet for coeliac disease, staple food availability was limited on the 2 weeks prior to 23rd march and for several weeks after”.

Additionally, one participant in IQ5 noted they were eating different brands of gluten free food available in smaller shops,

“I have coeliac disease and have been eating different brands of gluten free food during lockdown. I do not have a car so have had to use local stores. I’ve mostly shopped in small stores”

Participants in IQ1 had a 60% increased risk of experiencing food insecurity (RR = 1.6, CI: 1.1–2.4) compared to those not in IQ1 (Table 2).

### 3.2. Employment Type

Employment status during lockdown was associated with a relative likelihood of food insecurity. The self-employed, were significantly more likely to experience food insecurity than other groups (*p* = 0.037), whereas those in full-time employment were less vulnerable (Table 3). Whilst the numbers for several groups were too low to yield valid outcomes for Chi-square analysis (e.g., some expected values < 5) it is still worth noting that for those who recorded being unable to work either due to disability or being unemployed, the proportion experiencing food insecurity was very high.

The numbers for most groups changed during the lockdown with reductions in the number of self-employed (20%), part-time (12%), and full-time employed (17%) and increases in the number of unemployed (33%). There were also small increases in the number of people unable to work due to sickness or disability, homemakers/full-time parents, and retired. A large number were additionally placed on furlough.

### 3.3. Following Government Guidelines on Isolating and Shielding

The majority of participants (*n* = 442, 85.81%) at the time of the study were not self-isolating but following government guidance of social distancing. People not leaving their home because they were in the high-risk category accounted for 1.7% of the study population (*n* = 9). Individuals not leaving their home except to get essential items such as food and medicine accounted for 5.2% (*n* = 27). Whilst those not leaving the home because of living with someone vulnerable to the disease was 7.4% (*n* = 38) Participants who were living with someone vulnerable to COVID-19 were 1.88 (CI, 1.1–3.1) times more likely to report they were food insecure (*p* = 0.027; Table 4).

### 3.4. Concern for Food Availability

Over a quarter of all adults (27.8%) in this study said they were worried their household would not have enough food at the start of the first lockdown, of which a tenth indicated this was sometimes or often true (10.5%). Comparisons between food secure (*n* = 421) and food insecure (*n* = 94) indicated that 81.5% of food secure (*n* = 343) were not worried about running out of food and just 29.8% of food insecure participants (*n* = 28) were not worried.

### 3.5. Eating Preferred Food by Food Security Status and Income Quintile

Analysis by income quintiles found similar proportions of participants indicating that they were unable to eat the type of foods they preferred due to lack of food available (*p* = 0.624). Participants in IQ1 also reported a lack of money as a reason for not being able to eat the foods they preferred compared to IQ3 (*p* = 0.002), IQ4 (*p* = 0.016), and IQ5 (*p* = 0.009). Analysis by food security status indicated a greater proportion of food insecure participants (69.1%) were unable to eat the foods they preferred due to lack of foods available compared to food secure (36.1%; *p* < 0.001). Eating non-preferred foods because of a lack of money was true for some amongst the food insecure (12.8%) but not those who were food secure (0.0 %; *p* <0.001).

### 3.6. Differences in Household Income, Food Spend and Food Security Status

Most participants provided details of household income and food spend prior to and during COVID-19 (*n* = 468). There was an 11.1% difference in the median equivalised household income between food secure (*n* = 385) and food insecure (*n* = 85), with food secure households having on average £89.86 more per week during lockdown (*p* < 0.01). Median food spend per week during lockdown was similar for the food secure (£86.03 [£60.18–£115.02]) and food insecure (£89.32 [£57.13–£173.08]) per week; (*p* = 0.582). The proportion of income required for food spend in food secure respondents was 9.5% and food insecure 11.0% (Table 1).

### 3.7. Change in Food Spend Amount per Week during the First UK Lockdown by Food Security Status

Median food spend during the first UK lockdown was £86.51 per week. Households who increased their food spend did so on average by 44.0%, whilst households who decreased food spend did by 28.1% (Table 5). Food secure and food insecure households who increased food spend, did so by a similar proportion (43.7% and 46.7%, respectively). The percentage of income spent on food was numerically, but not significantly, greater for food insecure households compared to food secure (*p* = 0.151; Table 5). When households’ food spend remained the same during the lockdown, food insecure households spent a greater proportion of their income on food compared to food secure (*p* = 0.003; Table 5).

### 3.8. Proportion of Income Spent on Food and Relative Risk of Food Insecurity

The proportion of household income spent on food in the UK averages 10.8% [29]. Among those participants who spent 10% or more of their household income on food and non-alcoholic beverages, we found that the relative risk of food insecurity increased by 0.07-fold for every 1% increase in the proportion of income spent (Figure 1). When the percentage of equivalised household income spent on food exceeded 13%, the relative risk of food insecurity increased by 1.6 fold (CI 1.1–2.3; *p* = 0.016).

### 3.9. Shopping Habits

As shown previously, in person shopping was reduced during the lockdown and this occurred for food secure (*p* < 0.001) and food insecure participants (*p* < 0.001), however the frequency of shopping in person during lockdown was lower for food insecure (2.98 ± 1.54) compared to food secure (3.32 ± 1.35; *p* = 0.05). This coincided with a slight increase in the frequency of using click and collect in both food secure (*p* < 0.001) and food insecure groups (*p* = 0.017). Whilst there was not a significant difference in click and collect between food secure and food insecure groups during the lockdown, there was a clear trend, appearing higher in food insecure compared with food secure (*p* = 0.067). There was a reduction in the frequency of shopping at the big four UK supermarkets and discounter supermarkets for all households during the lockdown (*p* < 0.001).

### 3.10. Eating Habits

Participants were asked to self-report if they thought they had eaten less than, about the same, or more than their usual diet since the 23 March 2020. Those who were classified as food insecure (*n* = 94) were 2.1 (CI 1.4–3.0) times more likely to report they thought they were eating less than usual their usual diet, however, the majority of respondents felt that their diet was as healthy as it was prior to lockdown. Despite this, food secure households reported a lower proportion of the population decreasing their fresh fruit intake (15.8%) compared to food insecure (32.3%; *p* = 0.060) as well as fresh vegetable intake (11.8% vs. 35.1%, respectively; *p* = 0.001). Consumption of breakfast cereal remained the same for all (72.4% food secure and 61.3% food insecure).

### 3.11. Coping Strategies

We further asked participants to indicate if they had to employ any strategies to cope with not having enough food or money to buy food. Those who were food insecure more frequently relied on less preferred or less expensive foods than the food secure (*p* < 0.001). They were also 3.1 (CI 2.2–4.4) times more likely to reduce the quantity (*p* < 0.001) and 4.1 (CI 2.8–6.2) times more likely to reduce the variety of fruit and vegetables consumed (*p* < 0.001). Access to food banks was not reported by any respondents. Six out of the 8 people who reported purchasing food on credit were in the food secure category.

### 3.12. Diet Diaries, Nutrient Intakes, and Food Security Status

In total 6 males and 50 females completed a 3- or 4-day diet diary using the “Libro” app (Nutritics 2021). Because of the low numbers of males, only data for females was analysed for impact of food insecurity.

#### 3.12.1. Energy for All Participants

A non-plausible energy intake below the lower Goldberg cut off point was found in 11 participants. None exceeded the higher cut off point. One individual who was food insecure had EI: BMR below the Golberg limit, whilst 10 food secure participants were below the lower cut off.

#### 3.12.2. Energy Intakes and BMI

Energy intake of all females aged 19-65 years was 1751 ± 50.50 kcal. Seven food secure participants (12.5%) had an EI:BMR ratio below the lower cut-off point but were not excluded from the analysis. Food secure (*n* = 44) and insecure (*n* = 6) participants had similar energy intakes (1728 ± 53.85 kcal and 1924 ± 136.37 kcal, respectively). The BMI of food secure participants was within the healthy range (23.4) whilst those for the food insecure fell within the overweight category (28.5) although there was no significant difference between the two groups.

#### 3.12.3. Macronutrients

Carbohydrate intake provided slightly below the recommended 50% of energy for both food secure (44.0 ± 1.2%) and food insecure 42.9 ± 4.2% (*p* = 0.763), whilst energy from total fat exceeded the recommended 33% (36.5 ± 1.1% and 36.9 ± 3.2% food secure and insecure, respectively) but did not differ between the groups (*p* = 0.898). The 11% energy from saturated fat was similarly exceeded for both food secure and insecure (12.7 ± 0.7 % and 12.5 ± 1.5%, respectively; *p* = 0.933) as was protein intake (151.0% and 157.5% of DRV; *p* = 0.511). There were comparable intakes of fibre for both food secure and food insecure ~75% of the DRV (22.6 g [18.5–29.67] and 22.5 g [20.3–27.3], respectively; *p* = 0.965) and free sugar consumption was within the recommendations of ≤5% of energy for both groups (food secure—2.1% [1.1–5.0%] and food insecure—1.6% [0.7–5.1]; *p* = 0.456).

#### 3.12.4. Micronutrients—Vitamins

Intakes of most vitamins were similar, but B2 (riboflavin) was significantly negatively influenced by food insecurity (Figure 2A–D). Those who were classed as food insecure, consumed 36% less riboflavin than food secure individuals with values of 0.7 mg [0.6–1.1] and 1.1 mg [0.9–1.5], respectively (*p* = 0.030; Figure 2A).

Food secure females exceeded the RNI for folate (104.9% of the RNI) whilst food insecure intakes were lower than the RNI (82.6%), but no significant difference was found between groups (*p* = 0.222; Figure 2B).

Intakes of vitamin B12 were numerically very different between groups (2.5 μg [1.4–3.9] and 1.1 μg [0.5–2.5] for food secure and insecure, respectively), but this difference did not quite achieve significance (*p* = 0.057; Figure 2C). Vitamin A requirements for females 19+ in the UK are set at 600 µg d^-1^ Food insecure females had a low intake compared to the RNI (59.4% of the RNI) and consumed 51.6% less vitamin A compared to food secure individuals but again the difference, while considerable was not significant (*p* = 0.311; Figure 3D).

#### 3.12.5. Micronutrients—Minerals

Mineral intakes were not seen to vary significantly, however, consumption levels for several minerals were routinely low and also showed large numerical differences indicating trends towards lower intakes amongst the food insecure group (Figure 3A–E).

Iron intakes of females 19–49 years (*n* = 26) were low compared to the RNI with 96.2% consuming below the RNI (14.8 mg d^−1^) and 57.7% below the lower reference nutrient intake (LRNI; 8.0 mg d^−1^). Amongst those who were food insecure, iron intakes were 28.9% lower but not significantly so compared to food secure (5.65 ± 1.31 mg d^−1^ vs. 7.96 ± 0.70 mg d^−1^; *p* = 0.198; Figure 3A). Iron intakes were slightly higher in females over 50 years (8.37 ± 0.70 mg d^−1^) compared to 19–49 years and, as requirements are lower in this group, the proportion below the LRNI and RNI was reduced to 12.5% and 54.2%, respectively. Only two participants were food insecure in the over 50 group, so comparisons could not be made by food security status (Figure 3B).

Zinc from dietary sources was low compared to the RNI for the majority of females 19–65 years (median 84.2% [58.1–106.1] of RNI), with 70.0% below the RNI and 24.0% below the LRNI. This did not differ with food security status (*p* = 0.439; Figure 3C).

Iodine intakes were also low compared to the RNI (140µg d^−1^) for all (41.8% [23.3–66.7]) with 92.0% below the RNI and 54.0% below the LRNI (70 µg d^−1^). Whilst intake in food insecure females was 53.6% lower (46.1 µg d^−1^ [22.0–66.2]) compared to food secure (70.8 µg d^−1^ [34.3–99.6]) this was not significant (*p* = 0.257).

Two females aged 19–65 years in our study had a dietary intake of selenium above the RNI (60 µg d^−1^), with the remainder below the RNI and 74.0% were below the LRNI (40 µg d^−1^). No differences were detected between the food security groups (*p* = 1.000; Figure 3E).

## 4. Discussion

In this study, we found that lockdown destabilized the access to food and the perception of accessibility across all income domains, not exclusively those on lower incomes. However, those households with the lowest incomes did experience food insecurity more severely than those with higher incomes. We show that food insecurity is felt and feared in affluent groups as well as those on low incomes and that the anxiety around limited food availability drives behaviour change to ensure the security of personal acquisition. We find that the intake of a number of micronutrients is significantly below requirements and for vitamin B2 this is exacerbated by the presence of food insecurity

We found that, under lockdown conditions, having a low income, being self-employed or unemployed (for whatever reason), living with people vulnerable to disease and having children, greatly increased the likelihood of reduced food security. Notably these factors, in addition to limitations in availability, increased anxiety about household food provision. These findings are in line with previous work [30,31,32,33], but in this study, we specifically highlight the fact that these same factors are relevant to more affluent groups. Those categorised as food insecure were less able to eat preferred foods both due to availability and financial constraints and their shopping trips were reduced compared to food secure. It has been suggested that this may be because individuals with a higher level of education tend to have a more diverse diet from the outset [34]. There did not appear to be a movement to cheaper shops for the food insecure, but rather they appeared to try to replace with click and collect (albeit the association was not quite significant). We additionally saw clear trends indicating reduced intakes of important nutrients (e.g., riboflavin) in the food insecure group.

The nature of the participants, representing a relatively affluent cohort, highlighted an important aspect of the “food insecure” group. These people were, for the most part, very well educated and earning reasonable salaries, certainly to a level that would not be expected to be associated with food insecurity. Their responses, however, clearly indicated that they were either experiencing difficulties or had greater anxiety around their food security status, these findings align with a study in the US which found 19% of participants during the initial stages of COVID-19 (mid-March 2020) who had a very low food security status, had a high income (>$59,000 a year) whilst 21% with a graduate degree indicated they had a very low food security status [35]. A recent study suggested that a higher percentage of people reporting they were moderately or greatly affected during the pandemic in their shopping habits had postgraduate degrees compared with school, college or undergraduate degree holders [34]. Higher qualification levels were suggested to be associated with a more significant impact on food purchasing due to greater anxiety driving food purchases in the face of reduced variety of foods that are customarily available to them. This may have prompted our respondents to answer positively regarding worrying about running out of food, but this worry resulted in stockpiling and over-purchasing, thereby limiting further availability both for themselves and lower income households.

COVID-19 exposed fragilities in procuring food as the household income changes [9] but we found both food secure and food insecure households adjusted their shopping habits similarly. This suggested that, for this study population, the experience of food insecurity was primarily due to a lack of food availability rather than affordability, although households with the lowest incomes did indicate financial constraint was a factor in worrying about running out of food. Although income decreased in all groups, the proportional reduction was lower for food secure households, which may have afforded them the ability to still purchase food at the same level or potentially greater as a result of the reduction in extraneous costs (e.g., travel to work) [36]. Food budget modification was probably either limited or not required in this group. However, for other households, the higher percentage loss of income may have caused the food budget to be modified to pay for other household bills and costs. Limited availability of foods was the principal reason for the experience of food insecurity in this study and this was worsened by limited availability of supermarket delivery services, minimum spend and delivery cost associated with shopping online and restrictions on movement hampering access to shops [37]. During the first lockdown the infrastructure for click-and-collect and home delivery increased to cope with the demand [38,39], thereby diminishing the likelihood of limited food availability for wealthier households in subsequent lockdowns. This was probably of limited value to lower income households because of the constraint that the requirement for a minimum spend imposes [40].

The increase in food spend by wealthier households limited the availability of certain types of foods for others, such as core staple items of pasta, rice, and bread. The week before the first UK lockdown was imposed, increases in purchasing were observed across all social classes compared to the same time the previous year, however, households who were more affluent increased purchasing compared to less affluent groups [41].

Our data follows a similar trend to national data in that food insecurity was experienced at a greater level by younger individuals and those with a lower household income [42]. Furthermore, our results concur with studies researching the experience of food insecurity internationally, in that there was an increase in the experience of food insecurity during and after the initial lockdowns [43,44,45,46,47,48]. Households experiencing food insecurity spent a significantly greater proportion of their income on food. Furthermore, we found that the prevalence and severity of food insecurity was greatest for households when the proportion spent on food and non-alcoholic beverages exceeded 13% of equivalised household income. Engle’s law states that the proportion of income spent on food decreases as wealth increases [49]. Here, we further show that as the proportion of income spent on food increases, the risk of food insecurity increases directly with it (Figure 1). Within this population of relatively affluent individuals, we found that, above a percentage spend of 10% on food, the relative risk of food insecurity increased by 0.07 for every 1% increase in the proportion of income spent on food. It is unclear what the proportional increase in risk might be for lower income groups, but it can be envisaged that this is likely to be higher than in this case. With current food price increases, the proportional increase in food spend has already increased [50] latest data for commonly consumed food and drink items show prices have risen by 9.8% in the previous 12 months to June 2022 [50], placing a significant burden on all, but most severely those on low incomes.

Data from DEFRA showed that the proportion of household spend on food and non-alcoholic drinks in the UK in 2018/2019 was 10.6%, whilst for those in the lowest 20% income quintile this was 14.7% [51]. So, prior to COVID, these discrepancies were already apparent [52,53] and with the increased cost of living alongside the removal of the £20 uplift in universal credit [54], the proportion for the lowest income group will either be higher or have reached a threshold spend beyond which access to food banks or simply doing without food, becomes commonplace.

A Food Foundation survey during the COVID-19 Pandemic found that 14% of adults living with children reported experiencing moderate or severe food insecurity from March to September 2020 (~4 million people including 2.3 million children). The same survey found that 12% of adults living with children said they skipped meals, whilst 4% said that they had gone without food for a whole day because they could not afford or access food [55]. The UK Food Standards Agency (FSA) previously showed that adults living with children and particularly those on low incomes are more likely to experience food insecurity. Whilst most older people are food secure, around 10–20% of those aged 55 years and above experience some level of food insecurity [56].

We evaluated the dietary intakes of participants, albeit with a relatively small sample size. Riboflavin intakes were low compared to the RNI for food insecure females (77% of the RNI) with 67% below the LRNI. Riboflavin functions in a diverse array of redox reactions critical in cell metabolism via the cofactors, flavin mononucleotide (FMN) and flavin adenine dinucleotide (FAD), respectively. These cofactors act as important electron carriers in metabolism (e.g., succinate dehydrogenase) and in fatty acid oxidation. Recent evidence indicates that riboflavin deficiency can precipitate the development of anaemia and sub-optimal intakes are known to negatively impair iron utilization and hemoglobin synthesis [57]. Other research shows riboflavin deficiency causes metabolic dysregulation in animals [58] and can impact on the utilisation of other important B vitamins such as folate, vitamin B12 and vitamin B6 [59,60]. In addition to riboflavin, a general trend showing a reduction of vitamin B12 and vitamin B6 is also reported in participants, although these changes were not statistically significant. B vitamins, including B12 and B6 are known to be important in the metabolism of phospholipids and neurotransmitters and as such, deficiencies can cause haematological and neurological problems [61]. Moreover, an adequate intake of vitamins B1, B2, B12 and B3 are associated with a lower risk of NAFLD [62] and both B6 and B12 appear important in the metabolic pathway responsible for homocysteine metabolism, a marker for cardiovascular diseases [63]. Combined, it is feasible that sub-optimal levels of B vitamins, as reported in the current study, could predispose individuals to various health related problems.

Another trend observed in the current research was a reduction in vitamin A status in participants. Vitamin A has roles in growth, and in the prevention of night blindness [64] and reduced vitamin A status increases the risk of infection [65]. Marley et al. (2021) recently demonstrated that in patients replete or having abnormal levels of vitamin A have increased rates of inflammation and C-reactive protein [66]. This study pointing to an important role of vitamin A in mitigating rates of inflammation.

### 4.1. Limitations and Strengths of the Study

The data is in this study is largely self-reported, however, the tools used are validated methods. Although the tools in this study have been used in previous studies (e.g., HFIAS), they were not tested for the demographic in this study. Furthermore, the overall survey itself was not piloted prior to launch. This alongside the survey being completed online meant it was not possible to ascertain if the wording of the questions was fully understood by the participants. The population who participated in the study had above average levels of education and income and as such cannot be reflective of the general population. However, it does highlight food insecurity can be felt in population groups not typically thought to experience food insecurity. Indicating the prevalence of food insecurity in the UK is likely to be higher than reported and impacting a wider demographic of the UK population.

There have been limited studies measuring actual dietary intakes in food insecure groups, so by successfully utilizing a dietary monitoring app to capture food intake in geographically or socially isolated people across the age spectrum, as in this case, we have shown that this potentially represents a feasible means of obtaining dietary intake information in groups less physically accessible. We have additionally shown that the HFIAS tool, most usually employed to measure food insecurity as a result of financial/resource constraint, can usefully be employed to assess the impact of food availability in a western population. Using these approaches we show that groups at risk of food insecurity when faced with an unreliable food supply chain, can be identified. The results may aid policy maker’s decisions for the supply of funds/support for population groups at risk of the experience of food insecurity in the future.

### 4.2. Conclusions

Anyone can be food insecure or at least feel that they are. For those on higher incomes, “necessary” expenditure (e.g., debt servicing, elevated household bills) may prohibit prior freedoms of food purchasing if income is reduced or added financial burdens are placed upon them (children returning home, requirement to care for elderly relatives, energy, and mortgage cost increases). Additional drivers such as accessibility can further amplify the perception of personal food instability, all of which can result in disproportionate purchasing. These factors add to the burden of patent food insecurity amongst those with lower incomes as their ability to purchase what remains is compromised due to available choice and cost, with healthier foods in general being more expensive.

In this study, we found that despite mostly being in receipt of incomes approximating to the national average or higher, food insecurity was still experienced by nearly 20%. Those who experienced food insecurity had a lower household income (10.5% less) and were required to spend a much greater proportion of it (16% more) on food compared with food secure participants. However, food insecurity was predominantly driven by a lack of available food, although those in lower income groups indicated that financial constraint was a significant factor. Furthermore, when spend on food exceeded 13% of income the risk of experiencing food insecurity increased by 1.6 fold (*p* = 0.016).

Deficiency related diseases will be much more prevalent in those who are food insecure in the UK, and here we found that riboflavin intakes were 36% lower amongst food insecure compared to food secure individuals (*p* = 0.03). Whilst not significant, vitamin B12 intake was 56% lower and iodine, 53.6% lower in the food insecure, indicating a broader potential for deficiency in subgroups of food insecure participants. However, deficiency related diseases may still occur in people who are food secure, as deficiencies for specific nutrients, such as iron, are not uncommon (e.g., iron).

In summary, we observed a significant level of food insecurity within a population not typically considered at risk as >50% received a household income equivalent to or greater than the national average, resulting in specific nutritional intake deficits. The use of the proportion of income required to be spent on food has the potential to be an indicator for the risk of food insecurity and may help identify groups at risk when food spend equals or exceeds 13% of income.

## Figures and Tables

**Figure 1 nutrients-14-05078-f001:**
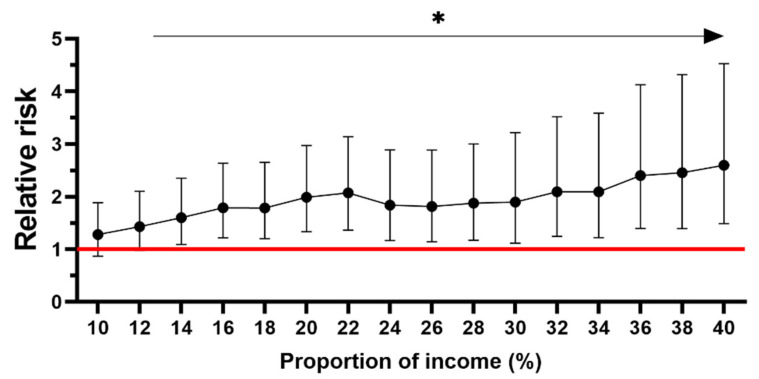
The relative risk of experiencing food insecurity when the proportion of income spent on food and non-alcoholic beverages exceeded 10% of income for food insecure vs. food secure. Results shown as relative risk with 95% confidence intervals. When household income spent on food exceeded 13%, the relative risk of food insecurity increased by 1.6 fold. Participants were included in the analysis if they provided details of their income and food spend (*n* = 468). * Significant at the *p* < 0.05 level (Pearson Chi-Square).

**Figure 2 nutrients-14-05078-f002:**
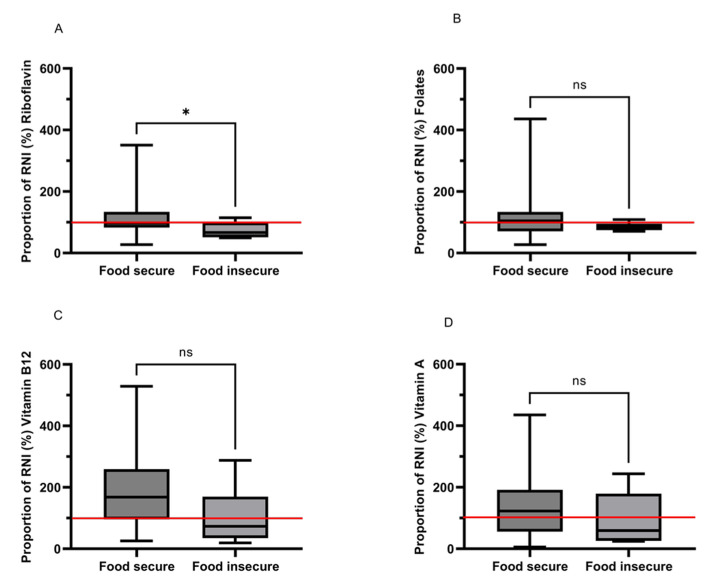
(**A**–**D**) The proportion of the Reference Nutrient Intake (RNI) met for vitamins amongst food secure (*n* = 44) and food insecure (*n* = 6) females aged 20–65 years. The red line indicates 100% of the RNI. * Significant at the *p* < 0.05. ns = non-significant.

**Figure 3 nutrients-14-05078-f003:**
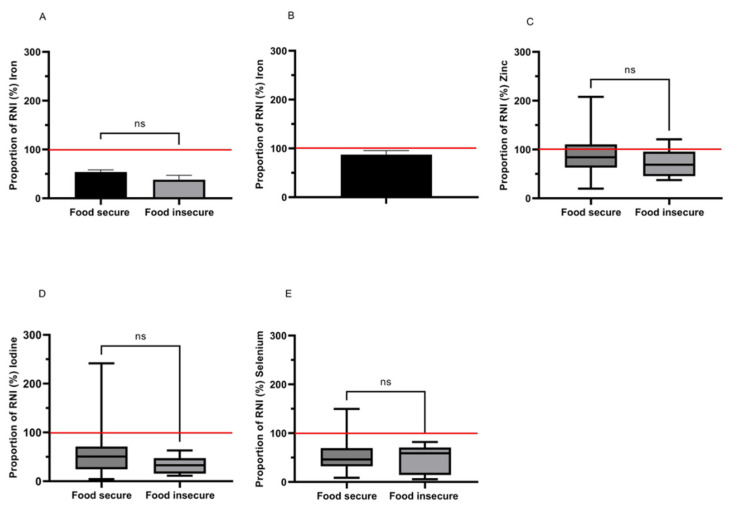
(**A**–**E**) The proportion of the Reference Nutrient Intake (RNI) met for minerals amongst food secure (*n* = 44) and food insecure (*n* = 6) females aged 20–65 years. The red line indicates 100% of the RNI. ns = non-significant.

**Table 1 nutrients-14-05078-t001:** Participant characteristics.

		**All**			**Food Secure**			**Food Insecure**			***p* Value**
		** *n* **	**Median**	**25th–75th Percentile**	** *n* **	**Median**	**25th–75th Percentile**	** *n* **	**Median**	**25th–75th Percentile**	
Age (years)		515	44.0	33–52	421	45.0	33–54	94	41.0	33–50	0.031 *
Height (m)	Male	77	1.80	1.75–1.85	62	1.80	1.75–1.85	15	1.80	1.78–1.85	0.395
Female		423	1.65	1.61–1.70	346	1.65	1.61–1.70	77	1.63	1.61–1.68	0.095
Missing		14	n/a	n/a	12	n/a	n/a	2	n/a	n/a	
Weight (kg)	Male	78	85.1	75.0–99.5	63	84.6	76.0–96.2	15	86.0	74.0–119.0	0.506
Female		412	66.0	59.9–78.0	336	66.0	60.0–77.4	76	67.3	58.2–81.8	0.837
Missing		24	n/a	n/a	21	n/a	n/a	3	n/a	n/a	
BMI	Male	76	26.1	23.7–30.1	61	25.9	23.7–29.8	15	27.1	23.1–31.1	0.681
Female		403	24.3	21.6–28.3	329	24.2	21.7–28.1	74	24.8	20.9–28.8	0.789
Missing		35	n/a	n/a	30	n/a	n/a	5	n/a	n/a	
Equivalised income per week (£)		470	853.69	551.69–1182.20	385	853.69	580.56–1243.71	85	763.83	288.69–1123.28	0.038 *
Food spend per week (£)		468	86.51	59.74–116.33	383	86.06	60.19–115.02	85	89.32	57.13–124.14	0.582
Proportion of income (%)		466	9.9	6.4–16.3	381	9.5	5.7–15.4	85	11.04	7.3–21.7	0.011 *
Household size		512	2.0	2.0–4.0	418	2.0	2.0–4.0	94	3.0	2.0–4.0	0.140
Sex		** *n* **	**(%)**		** *n* **	**(%)**		** *n* **	**(%)**		
	Male	79	(15.3)		64	(15.2)		15	(16.0)		0.861
Female		435	(84.5)		356	(84.6)		79	(84.0)		
Missing		1	(0.2)		1	(0.2)		n/a	n/a		
Income quintiles	1 (<£25,700.47)	98	(20.9)		72	(18.7)		26	(30.6)		0.117
2 (>£25,700.47)		90	(19.1)		75	(19.5)		15	(17.6)		
3 (>£39,643.18)		99	(21.1)		80	(20.8)		19	(22.4)		
4 (>£53,277.87)		84	(17.9)		73	(19)		11	(12.9)		
5 (>£75,503.02)		99	(21.1)		85	(22.1)		14	(16.5)		

Comparison between food secure and food insecure groups for Age (years), height (m), weight (kg), equivalised income, food spend, and the proportion of income spent on food as well as household size and body mass index (BMI) tested with MANN Whitney U (* indicates significance at the *p* < 0.05 level). Differences in the frequency of individuals represented in food secure and food insecure groups for sex and income quintile were tested with Pearson Chi Sq (* significant at the *p* < 0.05 level). n/a = not applicable.

**Table 2 nutrients-14-05078-t002:** Relative risk of food insecurity amongst income quintiles.

Income Quintile (per Year (£))	*n*	(%) ^ǂ^	RR	(CI)	*p* Value
1 (<£25,700.47)	26	(30.6)	1.6	(1.1–2.4)	0.015 *
2 (>£25,700.47)	15	(17.6)	0.9	(0.5–1.5)	0.697
3 (>£39,643.18)	19	(22.4)	1.1	(0.7–1.7)	0.747
4 (>£53,277.87)	11	(12.9)	0.7	(0.4–1.2)	0.190
5 (>£75,503.02)	14	(16.5)	0.7	(0.4–1.2)	0.251

RR = Relative Risk; CI = Confidence Interval; *n* Number of people who were food insecure; ^ǂ^ Percentage who were food insecure; * Significant at the *p* < 0.05 level (Pearson Chi Sq).

**Table 3 nutrients-14-05078-t003:** Relative risk of the experience of food insecurity by employment status.

Employment Status	Relative Risk (RR) of Food Insecurity If in Listed Employment before Lockdown RR (95% CI)						Relative Risk (RR) of Food Insecurity If in Listed Employment during Lockdown RR (95% CI)					
	*n*	(*n*) ^Ŧ^	(%) ^ǂ^	RR	CI	*p* Value	*n*	(*n*) ^Ŧ^	(%) ^ǂ^	RR	CI	*p* Value
Self-Employed	44	(12)	(27.3)	1.6	(0.9–2.6)	0.105	35	(11)	(31.4)	1.8	(1.1–3.1)	0.037 *
Part-time employment	93	(12)	(12.9)	0.7	(0.4–1.2)	0.140	82	(9)	(11.0)	0.6	(0.3–1.1)	0.063
Full-time employment	328	(55)	(16.8)	0.8	(0.6–1.2)	0.248	273	(45)	(16.5)	0.8	(0.6–1.2)	0.270
Unable to work due to disability	6	(3)	(50.0)	2.8	(1.2–6.4)	0.043 *^,¥^	8	(4)	(50.0)	2.8	(1.4–5.8)	0.019 *^¥^
Unable to work due to sickness	3	(3)	(100.0)	5.6	(4.7–6.8)	<0.001 *^,¥^	5	(3)	(60.0)	3.4	(1.6–7.0)	0.015 *^¥^
Unable to work as unemployed/seeking work	9	(5)	(55.6)	3.2	(1.7–5.8)	0.003 *^¥^	18	(8)	(44.4)	2.6	(1.5–4.5)	0.003 ^¥^
Homemaker/full-time parent	15	(3)	(25.0)	1.1	(0.4–3.1)	0.859 ^¥^	19	(5)	(26.3)	1.5	(0.7–3.2)	0.354 ^¥^
Furloughed worker	n/a	n/a	n/a	n/a	n/a	n/a	52	(10)	(19.2)	1.1	(0.6–1.9)	0.847
Prefer not to say	2	(1)	(50.0)	2.8	(0.7–11.2)	0.244 ^¥^	2	(1)	(50.0)	2.8	(0.7–11.2)	0.244 ^¥^

Association of employment type/status prior to and during lockdown and likelihood of food insecurity. Values for the relative risk of food insecurity (95% confidence intervals) are shown. The total number in each group (*n*) are also indicated. Values in columns headed **^Ŧ^** comprise the number of people in each group who were food insecure and values in the column headed ^ǂ^ are the proportion of the group to which they belong. 2 people recorded more than one reason for being unable to work. ^¥^ cells have expected count less than 5. * Significant at the *p* < 0.05 level (Pearson Chi-Square). n/a = not applicable

**Table 4 nutrients-14-05078-t004:** Relative risk of food insecurity when following government movement restriction guidelines.

QN		Relative Risk (RR) Of Food Insecurity If in Listed Government Guidelines					
		*n*	Ŧ (*n*)	ǂ (%)	RR	(CI)	*p* Value
1	Not self-isolating but following government guidance on social distancing	442	(72)	(16.3)	0.5	(0.4–0.8)	0.005 *
2	Self-isolating for 7 days, following symptoms	2	(1)	(50.0)	2.8	(0.7–11.2)	0.244
3	Self-isolating for longer than 7 days following symptoms, because you still have a temperature (above 37.8 °C)	1	(1)	(100.0)	5.5	(4.6–6.6)	0.034 *^,¥^
4	Self-isolating for LONGER than 14 days following symptoms in a member of your household, because YOU have developed symptoms during this time	0	(0)	(0.0)	n/a	n/a	0.636 ^¥^
5	Not leaving your home because you are at a VERY HIGH RISK of COVID-19 and have received a letter from the NHS (Shielding)	9	(1)	(11.1)	0.6	(0.1–3.9)	0.313 ^¥^
6	Not leaving your home except to get essential items such as food and medicine because you are at HIGH RISK of COVID-19, e.g., are 70 or older, pregnant, have diabetes, taking medication that can affect your immune system.	27	(9)	(50.0)	1.9	(1.1–3.4)	0.037 *^,¥^
7	Not leaving your home because someone in the household is more vulnerable to the virus (i.e., not high risk but at greater risk)	38	(12)	(31.6)	1.8	(1.1–3.1)	0.027 *
8	Status isolating or not Prefer not to say	3	(0)	(0.0)	n/a	n/a	0.412 ^¥^

QN = Question number; RR = Relative Risk; CI = Confidence interval; Ŧ Number of people who food were insecure; ǂ Percentage of employment type food insecure; ^¥^ cells have an expected count of less than 5; * Significant at the *p* < 0.05 level (Pearson Chi-Square). n/a = not applicable

**Table 5 nutrients-14-05078-t005:** Change in food spend from prior to, to during lockdown by food security status.

			Food Spend Change Groups			
			Increase	Stayed Same	Decrease	*p* Value
Total	*n*		174	209	83	
	Percentage of income spent on food (%)		11.5 ^a^	8.74 ^b^	9.2 ^b^	0.001 **
	Percentage of group (%)		(37.3)	(44.8)	(17.8)	
	Change in food spend (£)		+33.57	0	−30.31	
	EQVINC Weekly income (£)	Median	853.69	853.69	878.11	0.380
		25th	597.11	530.07	584.1	
		75th	1297.42	1182.2	1129.32	
Food secure	*n*		144	174	63	
	Percentage of income spent on food (%)		11.4 ^a^	7.9 ^b^	9.2^b^	<0.001 **
	Percentage of group (%)		(37.8)	(45.7)	(16.0)	
	Change in food spend (£)		+32.38	0	−28.88	
	EQVINC Weekly income (£)	Median	853.69	853.69	853.69	0.716
		25th	609.97	575.49	548.52	
		75th	1279.16	1269.49	1176.70	
Food insecure	*n*		30	35	20	
	Percentage of income spent on food (%)		14.3 ^a^	10.6 ^a,b^	8.4 ^b^	0.047 *
	Percentage of group (%)		(35.3)	(41.2)	(23.5)	
	Change in food spend (£)		+39.89	0	−35.9	
	EQVINC Weekly income (£)	Median	891.17	561.51	909.56	0.028 *
		25th	520.34	225.86	629.03	
		75th	1366.22	935.85	1024.57	
*p* Value	Food secure v food insecure percentage of income		0.151	0.003 *	0.866	
*p* Value	Food secure v food insecure Change in EQVINC Weekly income (£)		0.780	0.002 *	0.802	

Kruskal–Wallis used to test for differences in the proportion of income spent between the food spend groups for food secure and food insecure, Mann–Whitney U to test for difference between food secure and food insecure EQVINC = Equivalised household income, Letters differentiate significance across the categories * Significant at the *p* < 0.05, ** Significant at the *p* < 0.001.

## Data Availability

The data presented in this study are available on request from the corresponding author. The data are not publicly available due to privacy.

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
