# Peer review of "The Impact of the COVID-19 Pandemic on the Food Security of UK Adults Aged 20–65 Years (COVID-19 Food Security and Dietary Assessment Study)"

_nutrients, 2022, doi:10.3390/nu14235078_

Round 1

Reviewer 1 Report

Reviewer comments

Thank you for granting me the opportunity to review this interesting study. In this work, Thomas et al. investigated the impact of COVID-19 pandemic on the diet quality of UK adults aged 20-65 years. Kindly, find below my comments for your response.

Title: The authors should please consider tweaking the title a bit. Diet quality which in this case would be related to the macro and micronutrient composition of the participants’ part as captured in the “food diary” was not the only outcome of interest the authors assessed. Consequently, I suggest the authors add the food security bit to it which could capture “household food security”.

Abstract

Line 14-15: The authors should please indicate in “magnitude” the proportion of the participants that were food insecure and the “lower income” they had relative to their food secured group.

Line 18: The authors should kindly expand the abbreviation “RNI” as it is newly introduced there.

Line 19: please, indicate some of the other micronutrients.

Introduction

The introduction is very well written. The authors have provided an accurate background on the subject matter which makes it easier for the reader to follow.

Materials and Methods

My concern is with how the authors can guarantee that the participants understood the questionnaires that were administered online. It is essential to always pre-validate questionnaires administered for qualitative studies. This ensures the participants understand the questionnaires. In this work, since it was done online, perhaps the authors could potentially indicate that as a “Limitation?”

The author fails to highlight specific items on the instruments used for drafting the questionnaire. For example, Section 2.3 Food security measures, the authors should expand on some of items on the Household Food Insecurity Access Scale (HFIAS) and 2.4. Shopping habits and food spend “food shopping behaviours”. Highlighting the exact items captured on them could allow for easier adoption by others. This could result in increased repeatability of the work.

Line 153: Why the authors make the participants completion of the “4-day food diary” optional.

Line 158-159: Was the assessment of the micronutrients done by the software? If yes, please add that to the sentence. Regarding the participants who completed the food diary, did they have to download the app? How did the authors get to see the results of the micronutrient analysis as revealed by the app? Did the participants have to take a snap shot and send it to the authors or the authors were custodians of the app and thus could see the outcome of the micronutrient analysis?

Line 163: The authors should expand the abbreviation EI:BMR. Do the authors mean Energy Intake: Basal Metabolic Rate? If yes, could they indicate how the BMR was determined? Was it through Bioimpedance analysis? If yes, how did they ensure it was validated?

2.7. Data analysis.

Did the authors check for the normality of the data? If yes, they should kindly indicate how that was done.

Results

The authors indicated in the Materials and Methods section that some demographic data collected on participants include “ethnicity”. However, no results was presented on that in the “Results” section.

In Table 1, the authors have highlighted the rows with missing numbers with colour code. I think it will be great to maybe use “N/A” as “Not Applicable” for the “Median” and the “Percentile”.

Table must always stand alone. Consequently, the authors should expand the abbreviation BMI at the bottom of the Table. The authors should please explain at the bottom of Table 1 what the P values with * means in the last column of Table 1.

Discussion

The authors should please, add more references especially comparing the findings of the study to those carried out in other parts of the world.

Conclusion

Line 558: The authors should please contextualise the sentence to focus on the UK population as that was where the study was carried. The authors stratified the participants into “food secure” and “food insecure” groups. It will therefore be expedient to conclude on the objectives of the study using these groups of participants. Kindly, tweak it to meet the objectives.

Author Response

Response to reviewers' comments.

We would like to thank the reviewers for their thoughtful and constructive comments. We have addressed all points raised and made relevant changes where necessary. Changes are made with “Track changes” and appear as red text.

Reviewer 1 comments.

Thank you for granting me the opportunity to review this interesting study. In this work, Thomas et al. investigated the impact of COVID-19 pandemic on the diet quality of UK adults aged 20-65 years. Kindly, find below my comments for your response.

COMMENT.

Title: The authors should please consider tweaking the title a bit. Diet quality which in this case would be related to the macro and micronutrient composition of the participants’ part as captured in the “food diary” was not the only outcome of interest the authors assessed. Consequently, I suggest the authors add the food security bit to it which could capture “household food security”.

RESPONSE: We have removed diet quality and replaced with food security

COMMENT.

Abstract

Line 14-15: The authors should please indicate in “magnitude” the proportion of the participants that were food insecure and the “lower income” they had relative to their food secured group.

RESPONSE: The line has been modified to read “Food insecure (n = 85, 18.3%) had a 10.5% lower income and the money spent on food required a greater proportion of income.”

COMMENT.

Line 18: The authors should kindly expand the abbreviation “RNI” as it is newly introduced there.

Line 19: please, indicate some of the other micronutrients.

RESPONSE: This sentence has been modified to take account of the two comments above and now reads “Micronutrient intakes were low compared to the reference nutrient intake (RNI) for most females, with riboflavin being 36% lower in food insecure groups (P=0.03), whilst vitamin B12 was 56% lower (p = 0.057) and iodine 53.6% lower (p = 0.257) these were not significant.”

Introduction

The introduction is very well written. The authors have provided an accurate background on the subject matter which makes it easier for the reader to follow.

COMMENT.

Materials and Methods

My concern is with how the authors can guarantee that the participants understood the questionnaires that were administered online. It is essential to always pre-validate questionnaires administered for qualitative studies. This ensures the participants understand the questionnaires. In this work, since it was done online, perhaps the authors could potentially indicate that as a “Limitation?”

RESPONSE: We concede that is a limitation of our study and agree that it needs addressing. We have therefore included the following text in the discussion (Line 601) “Although the tools in this study have been used in previous studies (e.g., HFIAS), they were not tested for the demographic in this study. Furthermore, the overall survey itself was not piloted prior to launch. This alongside the survey being completed online meant it was not possible to ascertain if the wording of the questions was fully understood by the participants.”

COMMENT.

The author fails to highlight specific items on the instruments used for drafting the questionnaire. For example, Section 2.3 Food security measures, the authors should expand on some of items on the Household Food Insecurity Access Scale (HFIAS)…

RESPONSE: Apologies for this oversight. We have now modified the manuscript to include the following clarifications from line 142

“…The HFIAS assesses three different but related domains of food insecurity [25]. Positive responses across the domains indicate increasing severity of food insecurity experienced. We adapted the questions to evaluate if the experience of food insecurity was because of a lack of money or lack of food.

 Domain one is concerned with anxiety/worry of running out of food and asks the question (1) “Did you worry that your household would not have enough food”. Domain two includes three questions to assess if there was a reduction in the quality and variety of the food consumed. These questions asked (2) “were you or any household member not able to eat the kinds of foods you preferred because of a lack of money or lack of food available?”, (3) “Did you or any household member have to eat a limited variety of foods due to lack of money or food available?” and (4) “Did you or any household member have to eat same food that you really did not want to eat because of lack of money or lack of food available to obtain other types of food”.

The final domain asks five questions and is concerned with reduction in the quantity of food eaten and experience of hunger. The first asks (5) “Did you or any household member have to eat a smaller meal than you felt you needed because there was not enough food?”, and the second, (6) “Did you or any household member have to eat fewer meals in a day because there was not enough food?”. Additional questions ask (7) “Was there ever no food to eat of any kind in your household because of a lack of money or lack of food available to get food?”, (8) “Did you or any household member go to sleep at night hungry because there was not enough food?” and (9) “Did you or any household member go a whole day and night without eating anything because there was not enough food” [25].”

COMMENT.

…the authors should expand on some of items on the Household Food Insecurity Access Scale (HFIAS) and 2.4. Shopping habits and food spend “food shopping behaviours”. Highlighting the exact items captured on them could allow for easier adoption by others. This could result in increased repeatability of the work.

RESPONSE: The following detail has been included to address this oversight (from Line 169).

Participants were asked about their food shopping behaviours before and during the first UK national lockdown in reference to where food was purchased, how and how frequently (never less than once a month, 2-3 times per month, once a month, 2-4 times per week, 5-6 times per week, once a day, prefer not to say). The following question was asked with the following options for response “Which of following best describe where you purchased foods from? (Tick all that apply)”: 1) Shop at one of the UK “Big Four” supermarkets (Tesco, Sainsbury’s, Morrisons, Asda) 2) “In person”, 3) “home delivery”, 4) “Click and Collect” 5) Other supermarket (Aldi, Lidl, Iceland, Netto). 6) “Other supermarket “(Waitrose, Marks and Spencer), 7) smaller shops (e.g., Co-op, Tesco express, Sainsbury local), 8)” Corner Shops (e.g., Happy Shopper, 7-11, Spar), 9) “Markets”, 10) Local independents (e.g., butchers, bakers, green grocers). In addition, participants were asked whether they were self-isolating or shielding and their level of vulnerability. Individuals were asked about usual eating behaviours, dietary choices, perception of how food availability had changed, and how their diet had changed during the lockdown. Food spend was estimated for each household from the mid-point of the monetary bracket per week (<£46, £47-£69, £70-£90, £91-£115, £116-£138, £139-£161, >£162) selected by participants.

COMMENT.

Line 153: Why the authors make the participants completion of the “4-day food diary” optional.

RESPONSE: An important focus of the study was to assess the food security status of the population and wanted to recruit as many participants as possible in order to maximise confidence in our findings. We consequently chose not to make the completion of the food diary compulsory as we felt that this would greatly reduce the number of respondents.  We only had 56 people choose to complete the food diary, so feel justified in not imposing its compulsory completion as this supports our original concern.

COMMENT.

Line 158-159: Was the assessment of the micronutrients done by the software? If yes, please add that to the sentence.

RESPONSE: The software calculates total micronutrient intakes from food sources across the period of completion for the food diary. All additional analyses were carried out by the research team including achievement of adequacy for each participant. Values for RNIs were taken from the Dietary Reference Values for Food Energy and Nutrient for the United Kingdom. We have added the sentence “The macronutrient and micronutrient composition of each participant’s diet was calculated by the Nutritics software.” at line 192 to indicate that the software specifically only calculated nutrient profiles of the diet.

COMMENT.

Regarding the participants who completed the food diary, did they have to download the app?

RESPONSE: Participants details were inputted into Nutritics by the researchers and a link sent to participants for them to access the libro app which needed to be downloaded.

COMMENT.

How did the authors get to see the results of the micronutrient analysis as revealed by the app? Did the participants have to take a snapshot and send it to the authors, or the authors were custodians of the app and thus could see the outcome of the micronutrient analysis?

RESPONSE: The dietary information from each participant was recorded in the app (requires food searching by the participant and insertion into the food diary record). Data from the app is stored centrally by Nutritics and may be downloaded solely by the research team. Each participant’s results were combined into a single securely held database on the University of Nottingham server for analysis.

COMMENT.

Line 163: The authors should expand the abbreviation EI:BMR. Do the authors mean Energy Intake: Basal Metabolic Rate? If yes, could they indicate how the BMR was determined? Was it through Bioimpedance analysis? If yes, how did they ensure it was validated?

RESPONSE: The EI:BMR represents a calculation for estimates of both energy intake and basal metabolic rate. The sentence beginning at line 198 has been modified to make this clearer: “The plausibility of energy intake was assessed by estimating Energy Intake: Basal Metabolic Rate (EI:BMR) ratio using the Schofield equation to estimate BMR and applying the Goldberg upper and lower and cut-off points specific to physical activity level (PAL; Supplementary Tables 1 and 2) [26] [27].”

COMMENT.

2.7. Data analysis.

Did the authors check for the normality of the data? If yes, they should kindly indicate how that was done.

RESPONSE: We assessed normality of the data in SPSS with the Shapiro-Wilks test using values p < 0.05 to indicate non-normally distributed data. We have added the following sentence at line 205 “Normality of the data was assessed in SPSS using Shapiro -Wilks.”

COMMENT.

Results

The authors indicated in the Materials and Methods section that some demographic data collected on participants include “ethnicity”. However, no results was presented on that in the “Results” section.

RESPONSE: This has now been addressed by adding the following sentence at line 218: “The majority of participants in this study stated their ethnicity as white British (n = 422, (81.9%)), whilst 11.3% were white Irish or white other, 2.2% Asian, 0.4% white and black African or African, 0.2% Arab, 0.8% other and 0.2% preferred not to say. Two people did not provide details of their ethnicity.”

COMMENT.

In Table 1, the authors have highlighted the rows with missing numbers with colour code. I think it will be great to maybe use “N/A” as “Not Applicable” for the “Median” and the “Percentile”.

  • Highlighted rows removed and N/A inserted

COMMENT.

Table must always stand alone. Consequently, the authors should expand the abbreviation BMI at the bottom of the Table. The authors should please explain at the bottom of Table 1 what the P values with * means in the last column of Table 1.

RESPONSE: These clarifications have been added.

COMMENT.

Discussion

The authors should please, add more references especially comparing the findings of the study to those carried out in other parts of the world.

RESPONSE: We have added the following text at line 500: “…these findings align with a study in the US which found 19% of participants during the initial stages of COVID-19 (mid-March 2020) who had a very low food security status, had a high income (> $59,000 a year) whilst 21% with a graduate degree indicated they had a very low food security status [35].”

In addition we have added the following at line 540: “Furthermore, our results concur with studies researching the experience of food insecurity internationally, in that there was an increase in the experience of food insecurity during and after the initial lockdowns [43] [44] [45] [46] [47] [48].”

COMMENT.

Conclusion

Line 558: The authors should please contextualise the sentence to focus on the UK population as that was where the study was carried. The authors stratified the participants into “food secure” and “food insecure” groups. It will therefore be expedient to conclude on the objectives of the study using these groups of participants. Kindly, tweak it to meet the objectives.

RESPONSE: We have made modifications to address this point in the sentence beginning at line 632 (text highlighted red). The line now reads: “Deficiency related diseases will be much more prevalent in those who are food insecure in the UK, but may still exist in people who are food secure, being reflective of deficiencies for specific nutrients (e.g., riboflavin, iron, iodine) resulting in altered and less obviously diagnosed, but nonetheless significant pathologies.”

Reviewer 2 Report

The main aim of paper: ‘The impact of the COVID-19 pandemic on the diet quality of UK adults aged 20-65 years (COVID-19 Food Security and Dietary Assessment Study)’ was to assess the impact of social isolation and movement restriction on food availability and food security in UK adults during the first COVID-19 lockdown period.

From my point of view, the topic of the study is important and really interesting.In general, the procedure, the study design and statistical analyses are well organized.

However, I would like to indicate two minor suggestions for Authors:

(1)   You focused on the group of 20-65 years old people in the title of your paper, so correct this age on the page no 3 line 106.

(2)   Indicate some practical implications and strengths of your research.

Author Response

We would like to thank the reviewers for their thoughtful and constructive comments. We have addressed all points raised and made relevant changes where necessary. Changes are made with “Track changes” and appear as red text.

Reviewer 2

The main aim of paper: ‘The impact of the COVID-19 pandemic on the diet quality of UK adults aged 20-65 years (COVID-19 Food Security and Dietary Assessment Study)’ was to assess the impact of social isolation and movement restriction on food availability and food security in UK adults during the first COVID-19 lockdown period.

From my point of view, the topic of the study is important and really interesting.In general, the procedure, the study design and statistical analyses are well organized.

However, I would like to indicate two minor suggestions for Authors:

COMMENT.

  • You focused on the group of 20-65 years old people in the title of your paper, so correct this age on the page no 3 line 106.

RESPONSE: Apologies, this has been corrected (now line 107).

COMMENT.

  • Indicate some practical implications and strengths of your research.

RESPONSE: We have modified the Limitations and strengths section to include the following paragraph at line 611:

“ There have been limited studies measuring actual dietary intakes in food insecure groups, so by successfully utilizing a dietary monitoring app to capture food intake in ge-ographically or socially isolated people across the age spectrum, as in this case, we have shown that this potentially represents a feasible means of obtaining dietary intake infor-mation in groups less physically accessible. We have additionally shown that the HFIAS tool, most usually employed to measure food insecurity as a result of financial/resource constraint, can usefully be employed to assess the impact of food availability in a western population. Using these approaches we show that groups at risk of food insecurity when faced with an unreliable food supply chain, can be identified. The results may aid policy maker’s decisions for the supply of funds/support for population groups at risk of the ex-perience of food insecurity in the future.”

Round 2

Reviewer 1 Report

Thank you for making time to revise the work. I can see that there has been a great improvement in the quality of the content. My only comment is that the authors should please revise the content of the "Conclusion". I suggest that the authors address specifically the objectives of this work. In the present form, the writing appears like a general statements with no reference to this work. For example, the authors could add some of the findings captured in the "Abstract" to it as as to appear as "In this study, food insecure (n = 85, 18.3%) had a 10.5% lower income and the money spent on food required a greater proportion of income. Access to food was the biggest driver of food insecurity but monetary constraint was a factor for the lowest income group. The relative risk of food insecurity increased by 0.07-fold for every 1% increase in the proportion of income spent on food above 10%. Micronutrient intakes were low compared to the reference nutrient intake (RNI) for most females, with riboflavin being 36% lower in food insecure groups (P=0.03), whilst vitamin B12 was 56% lower (p = 0.057) and iodine 53.6% lower (p = 0.257) these were not significant."

Author Response

Response to reviewers' comments

Thank you for your feedback and constructive comments to improve the conclusion. We agree the conclusion lacked specificity and apologise for not addressing this in the first round of review. We have addressed the point raised and made the relevant changes. Changes are made with “track changes” and appear as red in the text.

Reviewer comments

Thank you for making time to revise the work. I can see that there has been a great improvement in the quality of the content. My only comment is that the authors should revise the content of the "Conclusion". I suggest that the authors address specifically the objectives of this work. In the present form, the writing appears like a general statements with no reference to this work. For example, the authors could add some of the findings captured in the "Abstract" to it as as to appear as "In this study, food insecure (n = 85, 18.3%) had a 10.5% lower income and the money spent on food required a greater proportion of income. Access to food was the biggest driver of food insecurity but monetary constraint was a factor for the lowest income group. The relative risk of food insecurity increased by 0.07-fold for every 1% increase in the proportion of income spent on food above 10%. Micronutrient intakes were low compared to the reference nutrient intake (RNI) for most females, with riboflavin being 36% lower in food insecure groups (P=0.03), whilst vitamin B12 was 56% lower (p = 0.057) and iodine 53.6% lower (p = 0.257) these were not significant."

Response: We have added the following to the conclusion starting at line 633

Anyone can be food insecure or at least feel that they are. For those on higher incomes, “necessary” expenditure (e.g., debt servicing, elevated household bills) may prohibit prior freedoms of food purchasing if income is reduced or added financial burdens are placed upon them (children returning home, requirement to care for elderly relatives, energy, and mortgage cost increases). Additional drivers such as accessibility can further amplify the perception of personal food instability, all of which can result in disproportionate purchasing. These factors add to the burden of patent food insecurity amongst those with lower incomes as their ability to purchase what remains is compromised due to available choice and cost, with healthier foods in general being more expensive.

In this study, we found that despite mostly being in receipt of incomes approximating to the national average or higher, food insecurity was still experienced by nearly 20%. Those who experienced food insecurity had a lower household income (10.5% less) and were required to spend a much greater proportion of it (16% more) on food compared with food secure participants. However, food insecurity was predominantly driven by a lack of available food, although those in lower income groups indicated that financial constraint was a significant factor. Furthermore, when spend on food exceeded 13% of income the risk of experiencing food insecurity increased by 1.6 fold (P = 0.016).

Deficiency related diseases will be much more prevalent in those who are food insecure in the UK, and here we found that riboflavin intakes were 36% lower amongst food insecure compared to food secure individuals (P = 0.03). Whilst not significant, vitamin B12 intake was 56% lower and iodine, 53.6% lower in the food insecure, indicating a broader potential for deficiency in subgroups of food insecure participants. However, deficiency related diseases may still occur in people who are food secure, as deficiencies for specific nutrients, such as iron, are not uncommon (e.g. iron).

In summary, we observed a significant level of food insecurity within a population not typically considered at risk as >50% received a household income equivalent to or greater than the national average, resulting in specific nutritional intake deficits. The use of the proportion of income required to be spent on food has the potential to be an indicator for the risk of food insecurity and may help identify groups at risk when food spend equals or exceeds 13% of income.
